# FROM NOISE TO SEMANTICS: BIT OPERATIONS DECOUPLE SPIKING NEURAL NETWORKS FOR ENTROPY OPTIMIZATION

## ABSTRACT

Although the binary spike transmission mechanism enables spiking neural networks (SNNs) to consume ultra-low power, SNNs have always struggled with suboptimal performance. This paper points out that SNN spike maps suffer from significant spike noise, which impairs object semantics and results in limited performance. To mitigate spike noise interference, we explored minimal bit operations for spike map decoupling. Specially, we show that the AND operation can ingeniously extract stable object semantics across timesteps, while the XOR operation separates out unstable spike noise. By approximating the original spike map through this decoupling, we propose an information bottleneck-based entropy optimization strategy that explicitly minimizes the conditional entropy of the object semantics while maximizing that of the spike noise. This dual entropy optimization strategy allows SNNs to ignore noise interference and learn the optimal semantic representation. To ensure efficiency, we replace the entire forward-backward propagation with a lightweight classifier to estimate conditional entropy, thereby introducing minimal training overhead. Extensive experiments have shown that our method significantly improves the performance of SNNs and offers superior generalization ability. In particular, our method can be seamlessly combined with others for flexible timestep inference and ultra-low latency early exit with a single training. This provides new insights into the efficient decoupling and optimization of SNNs.

## 1 INTRODUCTION

Compared to traditional artificial neural networks (ANNs) that convey continuous floating-point values, biomimetic spiking neural networks (SNNs) with iterative neuron dynamics convey information through sparse binary spikes, significantly reducing deployment power consumption and improving temporal properties Maass (1997); Yao et al. (2023); Ding et al. (2024). This has sparked widespread interest in SNNs, especially when combined with neuromorphic data represented by sparse events, which maximizes deployment power and speed gains Yao et al. (2024); Yang et al. (2024).

Despite their power consumption advantages, SNNs suffer from severe spike noise, which obscures critical object semantics and leads to suboptimal performance. We extensively visualized the spike maps of the optimized SNN in Fig. 1 and the Appendix A.5, and the results show that even a well-optimized SNN contains significant amount of semantically irrelevant spike noise. Specifically, since the spike map only contains 0 and 1 values, excessive 1-value spike noise has a more severe impact on the performance of SNNs than feature noise does on ANNs Bolya et al. (2023).

To mitigate the impact of spike noise, rather than relying on complex modules or technologies, we seek efficient solutions from minimal bit operations. Bit operations oriented to binary data can be efficiently implemented in hardware Bui et al. (2002); Kolesnikov & Schneider (2008), making them naturally suited for SNNs that also transmit binary spikes. The AND bit operation excels at identifying shared 1-value information ($\{1,1\}$) from binary input pairs. The XOR bit operation efficiently separates differences ($\{0,1\}$ or $\{1,0\}$) between two inputs. Exploiting these superior bit pattern separation properties, we show that AND and XOR can ingeniously separate stable object

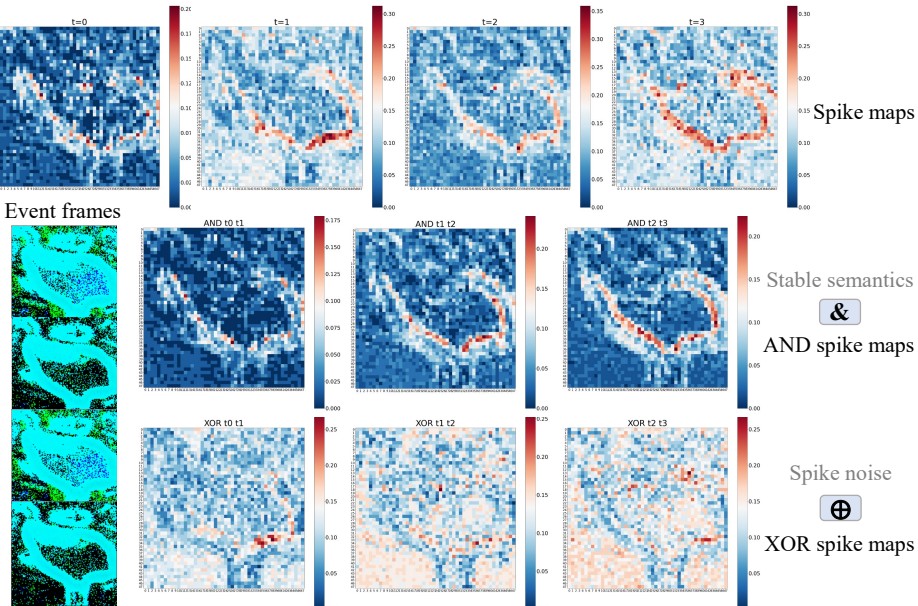

Figure 1: Visualization of spike maps generated by AND and XOR bit operations on CIFAR-10 DVS (The first layer of spiking VGG-9. See Appendix A.5 for more visualizations.). Compared to the original spike maps (first row), the AND spike maps (second row) extract well-defined object semantics, while the XOR spike maps (third row) capture spike noise.

semantics from unstable spike noise in spike maps. This allows us to decouple the spike features of SNNs efficiently and effectively without incurring significant additional overhead.

Furthermore, we utilize the information bottleneck principle Tishby & Zaslavsky (2015) to maximize/minimize the decoupled semantics/noise to improve the representation performance of the SNN. The information bottleneck principle aims to identify the minimum sufficient representation by retaining only the effective semantics and eliminating irrelevant noise. However, previous studies have shown that directly optimizing learning of the minimum sufficient representation is difficult. Instead, it is mainly learned through approximation methods, such as variational inference Wan et al. (2021); Lee et al. (2021); Li et al. (2025). Inspired by this, we take the original spike map in the SNN as the target and approximate it with object semantics and spike noise decoupled by AND and XOR operations. We convert the direct optimization of original spike maps into the minimization/maximization of object semantic/spike noise conditional entropy. This allows SNN to ignore noise interference and focus on the precise semantics. To maintain efficiency and avoid the overhead and gradient interference caused by direct forward-backward propagation of semantics and noise, we use a lightweight classifier to estimate conditional entropy. In this way, our method introduces negligible overhead during training, and its performance is theoretically guaranteed by the information bottleneck principle.

Through extensive experiments, we verified that the proposed method significantly improves the performance of low-latency SNNs. Additionally, our method can be easily combined with others and offers superior generalization ability. In combination with MPS Ding et al. (2025), we can flexible inference with reduced timesteps after a single training with a fixed timestep. This reduces the significant burden of repeatedly training SNNs adapted to different timestep constraints. With the early exit mechanism Li et al. (2023) to adjust the timestep for easy and hard samples, we can strike a great balance between performance and speed. For instance, on the CIFAR10-DVS dataset, we achieved an accuracy of 75.1% with an average timestep of only 2.28. This suggests that our method could advance the potential of other methods in the future. Our contribution can be summarized as follows:

- We show that minimal AND and XOR bit operations can efficiently decouple stable object semantics and unstable spike noise across timesteps.

- We propose an information bottleneck-based entropy optimization strategy to minimize/maximize the conditional entropy of object semantics/spike noise, prompting SNNs to learn optimal representations and mitigating noise interference.
- Extensive experiments demonstrate that our method significantly improves the performance of SNNs and enables flexible ultra-low latency inference with superior generalization.

## 2 BACKGROUND AND RELATED WORK

### 2.1 SPIKING NEURAL NETWORK

SNNs transmit sparse spikes between spiking neurons and have lower power consumption compared to traditional ANNs Yao et al. (2023). Moreover, spiking neurons iterate the charging, firing, and resetting processes over multiple timesteps, enabling the SNN to extract potential temporal features Ding et al. (2024). This paper employs the discrete leaky integrate-and-fire (LIF) neuron model Wu et al. (2018), which strikes a balance between biological plausibility and ease of implementation. The dynamics of the LIF neuron during a single timestep $t$ are as follows:

$$H_i^{l,t} = (1 - \frac{1}{\tau})H_i^{l,t-1} + I_i^{l,t}, \text{Charge} \tag{1}$$

$$S_i^{l,t} = \begin{cases} 1, & H_i^{l,t} \geq \vartheta \\ 0, & H_i^{l,t} < \vartheta \end{cases}, \text{Fire} \tag{2}$$

$$H_i^{l,t} = H_i^{l,t} - S_i^{l,t}\vartheta, \text{Reset} \tag{3}$$

where $H$ denotes the neuronal membrane potential, $\vartheta$ is the firing threshold, $S$ indicates the generated spike, and $l$ and $i$ denote the layer and neuron indices, respectively.

Note that the process of generating the spike (Eq. 2) is not differentiable. This hinders the training of SNNs by backpropagation. For this, methods have been developed to convert pre-trained ANNs to SNNs via rate or time mapping Li et al. (2021); Wang et al. (2025b) or to use surrogate gradients Wu et al. (2018); Anumasa et al. (2024); Zhou et al. (2024a) to circumvent the non-differentiability problem. In this paper, we use the rectangular surrogate function during backpropagation instead of Eq. 2 to compute the gradient for direct training, which preserves the temporal properties of the SNN compared to the conversion-based method Wu et al. (2018); Ding et al. (2024). The rectangular surrogate function can be expressed mathematically as follows:

$$\frac{\partial S_i^{l,t}}{\partial H_i^{l,t}} \approx \frac{\partial h(H_i^{l,t}, \vartheta)}{\partial H_i^{l,t}} = \frac{1}{a}\text{sign}(|H_i^{l,t} - \vartheta| < \frac{a}{2}), \tag{4}$$

where $a$ is the hyperparameter that controls the shape and defaults to 1.0.

Recent work has improved the performance of SNNs through various enhancements Huang et al. (2024); Zhang et al. (2024); Yu et al. (2025b). One notable enhancement is the spiking Transformer architecture, which uses attention mechanisms to focus more on semantics Yao et al. (2023); Zhou et al. (2024a). However, the conflict between semantic and spike noise has not been explicitly addressed. In this paper, we exploit bit operations to efficiently decouple spike semantics from noise, promoting the learning of optimal representations during training without sacrificing inference efficiency. Additionally, our method offers superior generalization ability and can be seamlessly integrated with other methods to further unleash the potential of SNNs.

### 2.2 INFORMATION BOTTLENECK PRINCIPLE

The information bottleneck was initially used for data compression Tishby et al. (2000). The objective is to compress the information source $X$ into a limited number of codewords $\hat{X}$ while preserving the information related to another signal $Y$. Then, this optimization objective is incorporated into deep learning, enabling the neural network to extract the most efficient features or approximate the *minimal sufficient statistics* with the most compact architecture Tishby & Zaslavsky (2015).

Formally, given a random variable $X$, which can be considered an input to a neural network, and a task-dependent variable $Y$, the learning objective, the information bottleneck seeks the optimal compressed representation (minimal sufficient statistics) $\hat{X} \in \hat{\mathcal{X}}$ that satisfies the following Lagrangian:

$$\min(I(X; \hat{X}) - \beta I(\hat{X}; Y)), \tag{5}$$

where $I(X; \hat{X})$ denotes the mutual information between $X$ and $\hat{X}$, and $\beta$ is the Lagrange multiplier.

Several methods exploit the information bottleneck principle to learn robust and efficient representations, and have made significant advancements in representation learning Wan et al. (2021); Lee et al. (2021); Li et al. (2025). In this paper, we take information bottlenecks as the theoretical foundation, utilize decoupled object semantics and spike noise to approximate the original spike maps, and optimize conditional entropy to enable the SNN to learn the minimum sufficient representation to improve performance.

## 3 METHOD

### 3.1 BIT OPERATIONS DECOUPLE SEMANTICS AND NOISE

The most distinguishing characteristic of SNNs, compared to ANNs that communicate with continuous activation values, is their delivery of single-bit spikes. This property renders SNNs inherently compatible with bit operations oriented to binary data, and enables its efficient execution on hardware Bui et al. (2002); Kolesnikov & Schneider (2008). This motivates us to explore the ingenious role of bit operations in SNNs.

In particular, in this paper we focus on AND and XOR bit operations. We denote the inputs for the bit operations as $x_1$ and $x_2$, and Table 1 shows the results generated by the AND and XOR operations for the four possible pairs of bits. Clearly, the AND operation extracts 1-value information ($\{1,1\}$) consistent between the two inputs while ignoring other interfering factors. In contrast, the XOR operation captures the $\{0,1\}$ and $\{1,0\}$ pairs inconsistent between the two inputs, i.e., where they differ. Consequently, the AND and XOR operations can distinguish between valuable (stable) and chaotic (unstable) information from the inputs, provided that the 1 values in inputs $x_1$ and $x_2$ accurately reflect the information. Fortunately, the multi-timestep spike maps in the SNN meet this prerequisite. Specifically, each timestep of the SNN can be regarded as a subnetwork, all of which are optimized to extract valuable information Ding et al. (2025). As shown in Fig. 1, spike maps of an optimized vanilla SNN with four timesteps reveal that 1-valued spikes constitute visual semantic features. Therefore, we believe that the distinctive characteristics of these two minimal bit operations can be leveraged to distinguish stable semantics from chaotic noise over timesteps in SNNs. We can extract $T-1$ object semantic/spike noise maps from $T$ original spike maps by considering two neighboring timestep spike maps as inputs to the bit operations.

Table 1: Input-output pairs for AND and XOR operations.

| Operation | Input $x_1$ | Input $x_2$ | Output |
|---|---|---|---|
| AND | 1 | 1 | 1 |
| | 1 | 0 | 0 |
| | 0 | 1 | 0 |
| | 0 | 0 | 0 |
| XOR | 1 | 1 | 0 |
| | 1 | 0 | 1 |
| | 0 | 1 | 1 |
| | 0 | 0 | 0 |

The second and third rows in Fig. 1 illustrate the results of employing AND and XOR operations on temporal neighboring spike maps, respectively. Compared with the original spike maps, the spike maps extracted by the AND operation exhibit clearer semantic contours and significantly reduced background noise. Conversely, spike maps extracted by the XOR operation prioritize background noise, resulting in a notable ambiguity in object semantics. This observation, along with the additional visualizations and analyses in the Appendix A.5, reveals consistent conclusions: **the AND operation can extract stable object semantics across timesteps, while the XOR operation captures significant spike noise**. Therefore, we can efficiently decouple spike maps using AND and XOR operations, without the complex modules and techniques required by ANNs Bolya et al. (2023); Littwin et al. (2024). Next, we will describe how to optimize SNNs with decoupled semantics and noise.

### 3.2 APPROXIMATE OPTIMIZATION OF MINIMAL SUFFICIENT REPRESENTATION IN SNNS

As shown in Eq. 5, the information bottleneck principle aims to find the minimum sufficient statistic $\hat{X}$ from the input $X$ that is relevant to the objective $Y$. In other words, it aims to minimize the mutual information $I(X; \hat{X})$ between $X$ and $\hat{X}$ while maximizing the mutual information $I(\hat{X}; Y)$ between $\hat{X}$ and $Y$. For discrete variables $\hat{X}$ and $Y$ (continuous variables are similar), we have:

$$I(\hat{X}; Y) = \sum_{\hat{x}} \sum_{y} p(\hat{x}, y) \log \frac{p(\hat{x}, y)}{p(\hat{x})p(y)} = \mathcal{H}(\hat{X}) - \mathcal{H}(\hat{X}|Y) = \mathcal{H}(Y) - \mathcal{H}(Y|\hat{X}), \quad (6)$$

where $\mathcal{H}(\cdot)$ is the information entropy. Since $Y$ is a deterministic target, such as the label of a classification task, $\mathcal{H}(Y)$ equals a constant $C$. Maximizing $I(\hat{X}; Y)$ can be converted into minimizing conditional entropy $\mathcal{H}(Y|\hat{X})$.

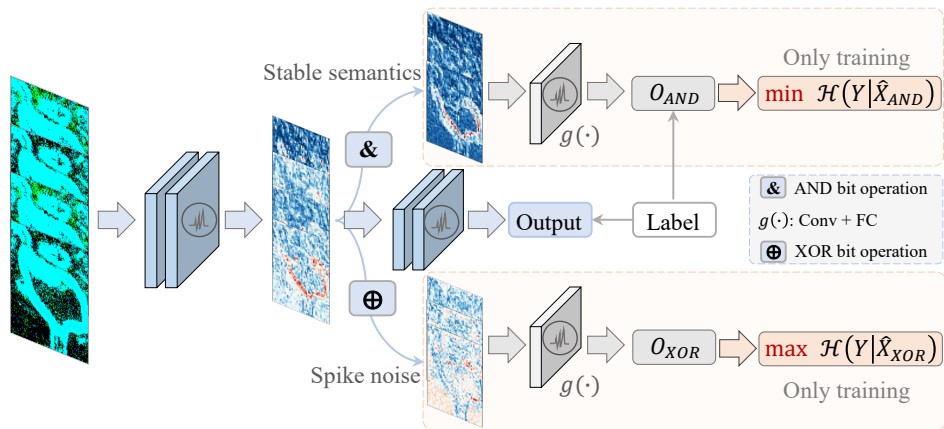

Figure 2: Overview of the proposed method. During training, we extract stable object semantics and spike noise using AND and XOR bit operations on the spike maps between adjacent timesteps. Minimizing the conditional entropy of stable semantic features while maximizing the conditional entropy of noisy features promotes learning the optimal representation.

In SNNs, we denote the spike sequence generated by spiking neurons as $\hat{X}$. In theory, we can minimize $\mathcal{H}(Y|\hat{X})$ directly towards the optimal information bottleneck, which is equivalent to minimizing the entropy of the SNN output. In fact, this is the standard SNN training process, with gradient descent driving the output close to the ground truth label. However, this simple training method does not maximize the performance of the SNN, as shown by the accuracies in Table 4. We attribute this to the excessive spike noise that cannot be eliminated, preventing the mutual information between $\hat{X}$ and $Y$ from being truly maximized. Therefore, instead of minimizing the conditional entropy $\mathcal{H}(Y|\hat{X})$ directly, we propose conducting feature separation in terms of AND and XOR operations to minimize/maximize the corresponding conditional entropy of the two, respectively.

Formally, we denote the spike map at the $t$-th timestep by $\hat{X}^t \in \mathbb{R}^{C \times H \times W}$, where $C$, $H$, and $W$ represent the channel, height and width, respectively, and $t \in \{0, 1, \cdots, T-1\}$. Stable object semantics $\hat{X}_{AND}$ and spike noise $\hat{X}_{XOR}$ can be efficiently extracted by applying AND and XOR operations between neighboring timesteps:

$$\hat{X}_{AND}^t = \text{AND}(\hat{X}^t, \hat{X}^{t+1}) = \hat{X}^t \& \hat{X}^{t+1}, \hat{X}_{XOR}^t = \text{XOR}(\hat{X}^t, \hat{X}^{t+1}) = \hat{X}^t \oplus \hat{X}^{t+1}, \quad (7)$$

where $\&$ and $\oplus$ denote the AND and XOR operations, respectively, and $t \in \{0, 1, \cdots, T-2\}$. Then, we approximate $\hat{X}$ with $\hat{X}_{AND}$ representing the object semantics and $\hat{X}_{XOR}$ representing the spike noise. Minimizing conditional entropy $\mathcal{H}(Y|\hat{X})$ is converted to minimizing the conditional entropy of valid semantic information and maximizing the conditional entropy of noise (theoretical analysis can be found in the Appendix A.2):

$$\min \mathcal{H}(Y|\hat{X}) \Rightarrow \min \mathcal{H}(Y|\hat{X}_{AND}) + \max \mathcal{H}(Y|\hat{X}_{XOR}). \quad (8)$$

However, directly computing $\mathcal{H}(Y|\hat{X}_{AND})$ and $\mathcal{H}(Y|\hat{X}_{XOR})$ requires forward propagating $\hat{X}_{AND}$ and $\hat{X}_{XOR}$ from the spike layer to the final classification layer to produce the output. This leads to significant training overhead and degraded performance due to gradient interference. To efficiently compute $\mathcal{H}(Y|\hat{X}_{AND})$ and $\mathcal{H}(Y|\hat{X}_{XOR})$, we introduce a lightweight classifier $g(\cdot)$ consisting of a $1 \times 1$ convolutional layer and a fully connected layer. The classifier $g(\cdot)$ receives inputs $\hat{X}_{AND}$ and $\hat{X}_{XOR}$ and produces outputs $O_{AND}$ and $O_{XOR}$:

$$O_{AND} = g(\hat{X}_{AND}), O_{XOR} = g(\hat{X}_{XOR}). \quad (9)$$

Then, we calculate the information entropy of $O_{AND}$ and $O_{XOR}$ to estimate the conditional entropy:

$$\mathcal{H}(O_{AND}) = -\sum_{k=1}^{K} p_{AND}^k \log(p_{AND}^k), \mathcal{H}(O_{XOR}) = -\sum_{k=1}^{K} p_{XOR}^k \log(p_{XOR}^k), \quad (10)$$

where $K$ is the class number, $p_{AND}$ and $p_{XOR}$ are the corresponding softmax probabilities for $O_{AND}$ and $O_{XOR}$.

### 3.3 OPTIMIZATION OBJECTIVES

It is imperative to ensure that the classifier $g(\cdot)$ produces task-relevant predictions; otherwise, merely decreasing/increasing $\mathcal{H}(O_{AND})/\mathcal{H}(O_{XOR})$ can result in training instability or even representation collapse. In light of this, the ground truth label $Y$ is used to guide $O_{AND}$ to promote $g(\cdot)$ toward the right optimization direction. Furthermore, guidance from label $Y$ to $O_{AND}$ facilitates propagation of the gradient in the early layers, mitigating the notorious gradient vanishing problem Ding et al. (2024). Thus, we formally define the loss term $\mathcal{L}_{AND}$ as follows:

$$\mathcal{L}_{AND} = \sum_{i=1}^{l} \mathcal{H}^i(O_{AND}) + \mathcal{L}_{CE}^i(O_{AND}, Y), \tag{11}$$

where $\mathcal{L}_{CE}$ is the cross-entropy loss, and $l$ denotes the number of layers for entropy optimization. To further reduce overhead, the lightweight classifier is separated into independent convolutional layers for each entropy optimization layer and a shared fully connected layer. The optimization of $\mathcal{L}_{AND}$ progressively reduces $\mathcal{H}(O_{AND})$ during training, and $\mathcal{H}(Y|\hat{X})$ is also approximately reduced, facilitating the learning of the optimal representation in the information bottleneck principle.

For $\hat{X}_{XOR}$, which signifies spike noise, we maximize its conditional entropy $\mathcal{H}(O_{XOR})$ as a loss term. This is equivalent to reducing the mutual information $I(\hat{X}_{XOR}; Y)$ between $\hat{X}_{XOR}$ and $Y$:

$$\min I(\hat{X}_{XOR}; Y) = \min(\mathcal{H}(Y) - \mathcal{H}(Y|\hat{X}_{XOR})) = \min(C - \mathcal{H}(Y|\hat{X}_{XOR}))$$
$$= \min(-\mathcal{H}(Y|\hat{X}_{XOR})) = \max \mathcal{H}(Y|\hat{X}_{XOR}) = \max \mathcal{H}(O_{XOR}), \tag{12}$$

where $C = \mathcal{H}(Y)$ is a constant. Reducing the mutual information $I(\hat{X}_{XOR}; Y)$ can degrade the amount of effective information in $\hat{X}_{XOR}$ so that the SNN focuses more on object semantics rather than spike noise. It is worth noting that when optimizing the conditional entropy $\mathcal{H}(O_{XOR})$ alone, it is necessary to ensure the optimization direction of the classifier $g(\cdot)$ with $\mathcal{L}_{CE}(O_{AND}, Y)$, but this is not necessary when optimizing in conjunction with $\mathcal{L}_{AND}$. Accordingly, the loss term $\mathcal{L}_{XOR}$ is formally defined as follows:

$$\mathcal{L}_{XOR} = \begin{cases} \sum_{i=1}^{l} -\mathcal{H}^i(O_{XOR}), \text{if with } \mathcal{L}_{AND} \\ \sum_{i=1}^{l} (-\mathcal{H}^i(O_{XOR}) + \mathcal{L}_{CE}^i(O_{AND}, Y)), \text{else} \end{cases}. \tag{13}$$

During training, $\mathcal{L}_{AND}$ and $\mathcal{L}_{XOR}$ are integrated with the original classification loss $\mathcal{L}_{CE}(O, Y)$ to co-optimize the parameters:

$$\mathcal{L}_{total} = \mathcal{L}_{CE}(O, Y) + \gamma_{AND}\mathcal{L}_{AND} + \gamma_{XOR}\mathcal{L}_{XOR}, \tag{14}$$

where $O$ is the original output of the SNN. $\gamma_{AND}$ and $\gamma_{XOR}$ are balance coefficients whose influence is analyzed in Table 3. Fig. 2 illustrates the overview of our method, and Alg. 1 outlines the specific optimization steps.

It is worth noting that the bit operations in Eq. 7 cannot be used directly in backpropagation. To this end, we approximate the gradient of the bit operations in backpropagation using additional surrogate functions. During forward propagation, the numerical results of $x_1 \& x_2$ and $x_1 \oplus x_2$ are equal to those of $x_1 x_2$ and $x_1 + x_2 - 2x_1 x_2$, respectively. Given this, we approximate the gradient of the AND and XOR operations in backpropagation as follows:

$$\frac{\partial(x_1 \& x_2)}{\partial x_1} \approx \frac{\partial(x_1 x_2)}{\partial x_1} = x_2, \frac{\partial(x_1 \oplus x_2)}{\partial x_1} \approx \frac{\partial(x_1 + x_2 - 2x_1 x_2)}{\partial x_1} = 1 - 2x_2. \tag{15}$$

With this, our method can directly train SNNs with backpropagation. Additionally, minimal bit operations and the lightweight classifier result in only negligible training overhead (For the ResNet-19 in Table 2, only 3.62% of the parameters and 2.62% of the FLOPs training overhead were produced.), prompting the SNN to explicitly focus on object semantics and degrade spike noise, significantly improving representation performance. It is worth noting that our method is identical to the vanilla

Table 2: Comparative results of training overhead. VGGSNN on CIFAR10-DVS with $T = 10$ and ResNet-19 on CIFAR10 with $T = 4$.

| Architecture | Method | Parameters ($\times 10^6$) | FLOPs ($\times 10^9$) |
|---|---|---|---|
| VGGSNN | Vanilla | 9.27 | 13.65 |
| | Ours | 10.00 (+7.87%) | 14.22 (+4.18%) |
| ResNet-19 | Vanilla | 12.70 | 9.17 |
| | Ours | 13.16 (+3.62%) | 9.41 (+2.62%) |

method during inference, thus introducing no additional inference overhead and maintaining universality across architectures.

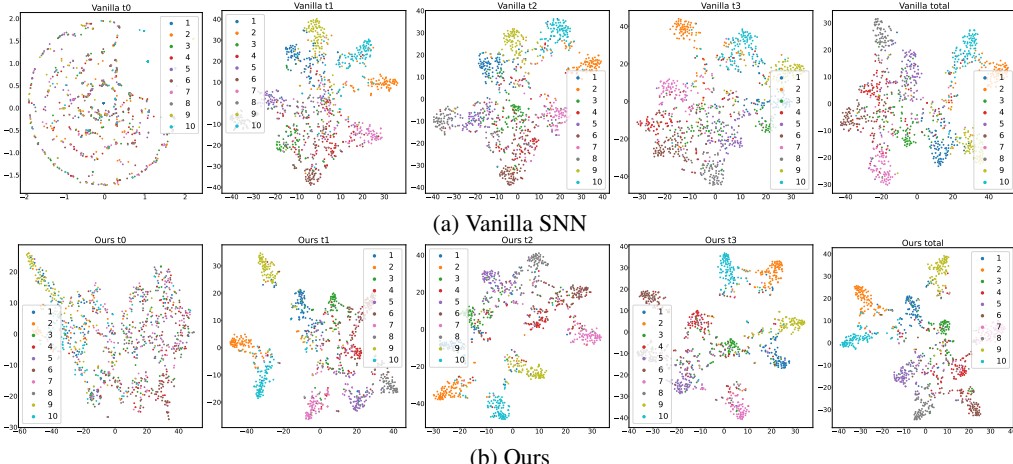

(a) Vanilla SNN

(b) Ours

Figure 3: The t-SNE visualization of the distribution of each timestep and total output. (a) The output distribution of the vanilla SNN is severely confused between classes, resulting in poor performance. (b) Our method significantly improves output discriminability by optimizing object semantics and reducing noise interference, as reflected by larger inter-class differences.

## 4 EXPERIMENTS

We conducted experiments on both neuromorphic (CIFAR10-DVS Li et al. (2017), DVS-Gesture Amir et al. (2017), and N-Caltech101 Orchard et al. (2015)) and static (CIFAR10/100 and ImageNet) datasets. However, our primary focus was on neuromorphic recognition performance with low latency because it fully exploits the speed and power consumption advantages of SNNs Yao et al. (2024); Yang et al. (2024). To validate the universality of the proposed method, experiments were conducted using VGG, ResNet, and spike Transformer architectures. The default timestep is set to 4, and the experimental details are provided in Appendix A.3.

### 4.1 ABLATION STUDY

The ablation study was conducted using VGG-9 on CIFAR10-DVS and DVS-Gesture. To reduce the impact of randomness, we report the average results of three trials.

**Influence of balance coefficients.** We verified the influence of $\gamma_{AND}$ and $\gamma_{XOR}$ in Table 3. For each experiment, one coefficient is kept at 1.0 while the other is adjusted. As shown in Table 3, the balance coefficient can cause performance fluctuations. However, our method achieves consistent, satisfactory performance without significant degradation. This also indicates that our method does not require deliberate adjustment of hyperparameters and can be plug-and-play to improve performance.

Table 3: Influence of balance coefficients $\gamma_{AND}$ and $\gamma_{XOR}$.

| $\gamma_{AND}$ | 0.1 | 0.3 | 0.5 | 0.8 | 1.0 | 1.3 | 1.5 |
|---|---|---|---|---|---|---|---|
| CIFAR10-DVS | 77.50 | 77.43 | 76.97 | 77.13 | **78.07** | 77.03 | 76.93 |
| DVS-Gesture | 92.59 | 93.52 | 93.40 | **95.14** | 94.10 | 94.21 | 94.56 |

| $\gamma_{XOR}$ | 0.1 | 0.3 | 0.5 | 0.8 | 1.0 | 1.3 | 1.5 |
|---|---|---|---|---|---|---|---|
| CIFAR10-DVS | 77.20 | 77.33 | 77.03 | 76.60 | **78.07** | 77.20 | 77.50 |
| DVS-Gesture | 94.79 | 93.17 | 94.91 | 93.98 | 94.10 | 94.33 | 94.10 |

ment of hyperparameters and can be plug-and-play to improve performance. We use the best balance coefficient value from Table 3 in subsequent experiments; otherwise, the default value is 1.0.

**Compared to the baseline.** Table 4 shows the ablation results compared to the baseline. The results show that using $\mathcal{L}_{AND}$ and $\mathcal{L}_{XOR}$ separately both produced significant performance gains, and using them together further improved performance. Notably, performance on DVS-Gesture increased by up to 7.99%, significantly improving the recognition of neuromorphic objects using low-latency SNNs.

Table 4: Results of ablation studies.

| $\mathcal{L}_{AND}$ | $\mathcal{L}_{XOR}$ | CIFAR10-DVS | DVS-Gesture |
|---|---|---|---|
| | | 73.23 | 87.15 |
| ✓ | | $76.93_{+3.70}$ | $94.10_{+6.95}$ |
| | ✓ | $76.83_{+3.60}$ | $91.67_{+4.52}$ |
| ✓ | ✓ | $\mathbf{78.07}_{+4.84}$ | $\mathbf{95.14}_{+7.99}$ |

Table 5: Comparative results with other methods. $^\dagger$: knowledge transfer from static data.

| Dataset | Method | Architecture | $T\downarrow$ | Accuracy (%) $\uparrow$ |
|---|---|---|---|---|
| CIFAR10-DVS | STAA-SNN$^{CVPR'25}$ (Zhang et al., 2025) | VGG-13 | 16 | 82.10 |
| | TS-SNN$^{ICML'25}$ (Yu et al., 2025c) | ResNet-20 | 10 | 83.80 |
| | SLT$^{AAAI'24}$ (Anumasa et al., 2024) | VGGSNN | 10 | 81.46 |
| | DeepTAGE$^{ICLR'25}$ (Liu et al., 2025) | VGG-11 | 10 | 81.23 |
| | ReverB-SNN$^{ICML'25}$ (Guo et al., 2025) | ResNet20 | 10 | 78.10 |
| | SSNN$^{AAAI'24}$ (Ding et al., 2024) | VGG-9 | 5 | 73.63 |
| | MPS$^{ICLR'25}$ (Ding et al., 2025) | VGG-9 | 5 | 76.77 |
| | | VGGSNN | 4 | 83.20 |
| | **Ours** | VGG-9 | 4 | **78.07** |
| | | VGGSNN | 4 | **84.30** |
| | | VGGSNN | 10 | **86.60** |
| | QKFormer$^{NeurIPS'24}$ (Zhou et al., 2024a) | QKFormer | 16 | 84.00 |
| | SNN-ViT$^{ICLR'25}$ (Wang et al., 2025a) | SNN-ViT | 16 | 82.30 |
| | SWformer$^{ECCV'24}$ (Fang et al., 2024) | SWformer | 10 | 82.90 |
| | SEMM$^{NeurIPS'24}$ (Zhou et al., 2024b) | Spikingformer | 10 | 80.70 |
| | SpikingResformer$^{CVPR'24}$ (Shi et al., 2024) | SpikingResformer | 10 | 81.50 |
| | **Ours** | QKFormer | 8 | **84.00** |
| | | | 16 | **85.00** |
| DVS-Gesture | MPS$^{ICLR'25}$ (Ding et al., 2025) | VGG-9 | 5 | 93.23 |
| | SSNN$^{AAAI'24}$ (Ding et al., 2024) | VGG-9 | 5 | 90.74 |
| | CLIF$^{ICML'24}$ (Huang et al., 2024) | VGG-9 | 4 | 89.58 |
| | TAB$^{ICLR'24}$ (Jiang et al., 2024) | VGG-9 | 4 | 87.50 |
| | SLT$^{AAAI'24}$ (Anumasa et al., 2024) | VGG-9 | 4 | 88.19 |
| | **Ours** | VGG-9 | 4 | **95.14** |
| | QKFormer$^{NeurIPS'24}$ (Zhou et al., 2024a) | QKFormer | 16 | 98.60 |
| | SpikingResformer$^{CVPR'24}$ (Shi et al., 2024) | SpikingResformer | 16 | 98.60 |
| | SEMM$^{NeurIPS'24}$ (Zhou et al., 2024b) | Spikingformer | 10 | 96.88 |
| | MPS$^{ICLR'25}$ (Ding et al., 2025) | SpikingResformer | 5 | 94.44 |
| | QKFormer$^{NeurIPS'24}$ (Zhou et al., 2024a) | QKFormer | 4 | 93.75 |
| | **Ours** | QKFormer | 4 | **95.49** |
| | | | 8 | **97.22** |
| | | | 16 | **98.61** |
| N-Caltech101 | SWformer$^{ECCV'24}$ (Fang et al., 2024) | SWformer | 10 | 88.45 |
| | TMC$^{ICML'25}$ (Yan et al., 2025) | VGGSNN | 10 | 86.03 |
| | IMP+TET-S$^{NeurIPS'24}$ (Shen et al., 2024b) | VGGSNN | 10 | 85.01 |
| | TKS$^{IEEE\ TAI'24}$ (Dong et al., 2024) | VGGSNN | 10 | 84.10 |
| | TIM$^{IJCAI'24}$ (Shen et al., 2024c) | Spikformer | 10 | 79.00 |
| | Knowledge-Transfer$^{AAAI'24}$ (He et al., 2024) | VGGSNN | 10 | 93.18$^\dagger$ |
| | SSNN$^{AAAI'24}$ (Ding et al., 2024) | VGG-9 | 5 | 77.97 |
| | MPS$^{ICLR'25}$ (Ding et al., 2025) | VGG-9 | 5 | 82.71 |
| | | VGGSNN | 10 | 93.68$^\dagger$ |
| | **Ours** | VGG-9 | 4 | **83.81** |
| | | VGGSNN | 10 | **94.60$^\dagger$** |

## 4.2 VISUALIZATION

To further investigate the effects of maximizing semantics while minimizing noise, Fig. 3 shows t-SNE visualizations of each timestep and total output for both the vanilla SNN and our method on CIFAR10-DVS. The results show that the output distribution of the vanilla SNN is chaotic, and severe interclass confusion causes poor performance. In contrast, our method optimizes object semantics and degrades noise, producing output distributions that are interclass separable and intraclass compact. This discriminative output distribution enables our method to achieve superior performance.

## 4.3 COMPARISON WITH OTHER METHODS

**Neuromorphic Dataset.** As shown in Table 5, our VGGSNN and QKFormer models achieved accuracies of 86.60% and 85.00%, respectively, on the CIFAR10-DVS dataset, significantly outperforming other methods. Especially at low latency, our VGGSNN achieves an accuracy of 84.30% with only 4 timesteps, surpassing the performance of other methods with more timesteps. Similarly, on DVS-Gesture and N-Caltech101, our method consistently outperformed other methods.

**Static Dataset.** As shown in Table 6, our ResNet-19 achieved accuracies of 96.68% and 81.69% on CIFAR10 and CIFAR100, respectively. Using the QKFormer architecture Zhou et al. (2024a), we also achieved better performance than the original method. Table 11 shows the comparative results achieved using ResNet-18 on ImageNet, where our method also outperforms other methods. This validates the effectiveness of our method on static datasets.

Table 6: Comparative results on static CIFAR10 and CIFAR100 datasets.

| Dataset | Method | Architecture | $T \downarrow$ | Accuracy (%) $\uparrow$ |
|---|---|---|---|---|
| CIFAR10 | QKFormer$^{NeurIPS'24}$ (Zhou et al., 2024a) | QKFormer | 4 | 96.18 |
| | SNN-ViT$^{ICLR'25}$ (Wang et al., 2025a) | SNN-ViT | 4 | 96.10 |
| | RateBP$^{NeurIPS'24}$ Yu et al. (2024) | ResNet-19 | 4 | 96.26 |
| | SLT$^{AAAI'24}$ (Anumasa et al., 2024) | ResNet-19 | 4 | 95.18 |
| | LSG$^{IJCAI'23}$ Lian et al. (2023) | ResNet-19 | 4 | 95.17 |
| | RateBP$^{NeurIPS'24}$ Yu et al. (2024) | ResNet-18 | 4 | 95.61 |
| | TWKD$^{ICML'25}$ Yu et al. (2025a) | ResNet-18 | 4 | 95.57 |
| | SLTT$^{ICCV'23}$ Meng et al. (2023) | ResNet-18 | 6 | 94.59 |
| | KDSNN$^{CVPR'23}$ Xu et al. (2023b) | ResNet-18 | 4 | 93.41 |
| | **Ours** | ResNet-18 | 4 | **95.84** |
| | | ResNet-19 | 4 | **96.68** |
| | | QKFormer | 4 | **96.42** |
| CIFAR100 | QKFormer$^{NeurIPS'24}$ (Zhou et al., 2024a) | QKFormer | 4 | 81.15 |
| | SNN-ViT$^{ICLR'25}$ (Wang et al., 2025a) | SNN-ViT | 4 | 80.10 |
| | DeepTAGE$^{ICLR'25}$ (Liu et al., 2025) | ResNet-19 | 4 | 81.39 |
| | RateBP$^{NeurIPS'24}$ Yu et al. (2024) | ResNet-19 | 4 | 80.71 |
| | TS-SNN$^{ICML'25}$ (Yu et al., 2025c) | ResNet-19 | 2 | 80.28 |
| | ReverB-SNN$^{ICML'25}$ (Guo et al., 2025) | ResNet-19 | 2 | 78.46 |
| | TMC$^{ICML'25}$ (Yan et al., 2025) | ResNet-19 | 4 | 77.52 |
| | SLT$^{AAAI'24}$ (Anumasa et al., 2024) | ResNet-19 | 4 | 75.01 |
| | RateBP$^{NeurIPS'24}$ Yu et al. (2024) | ResNet-18 | 4 | 78.26 |
| | SLTT$^{ICCV'23}$ Meng et al. (2023) | ResNet-18 | 6 | 74.67 |
| | **Ours** | ResNet-18 | 4 | **78.08** |
| | | ResNet-19 | 4 | **81.69** |
| | | QKFormer | 4 | **81.39** |

## 4.4 GENERALIZATION AND ULTRA-LOW LATENCY INFERENCE

Furthermore, we combine our method with existing methods on CIFAR10-DVS to verify its generalizability and potential for ultra-low latency inference. On the one hand, we collaborate with SEENN Li et al. (2023) to use the early exit mechanism to adopt different timesteps for difficult and easy samples to reduce inference latency. As shown in Table 7, our method combined with the early exit mechanism achieves an excellent balance of performance and speed. In particular, we achieved an accuracy of 75.1% with an average timestep of only 2.28 when the exit threshold was set to 0.5. On the other hand, we collaborate with

Table 7: The results of ultra low-latency inference with the early exit mechanism. $\vartheta_{exit}$ is the threshold for early exit.

| Method | $T \downarrow$ | Acc. (%) $\uparrow$ |
|---|---|---|
| Vanilla SNN | 4 | 72.9 |
| **Ours** | 4 | **78.2** |
| Early exit ($\vartheta_{exit} = 0.7$) | 2.86 | 72.2 |
| **Ours**+Early exit ($\vartheta_{exit} = 0.7$) | 2.61 | **77.3** |
| Early exit ($\vartheta_{exit} = 0.5$) | 2.56 | 71.6 |
| **Ours**+Early exit ($\vartheta_{exit} = 0.5$) | 2.28 | **75.1** |
| Early exit ($\vartheta_{exit} = 0.3$) | 2.21 | 68.3 |
| **Ours**+Early exit ($\vartheta_{exit} = 0.3$) | 1.98 | **72.0** |

MPS Ding et al. (2025) to explore reduced inference latency by training at a fixed 4 timestep. Table 8 indicates that our method effectively improves the performance in reduced inference latency, and the effect is even more significant when combined with MPS. This avoids the huge overhead of retraining the SNN to adapt to different timesteps.

## 5 CONCLUSION

In this paper, we decouple the object semantics and spike noise in SNNs by minimal AND and XOR bit operations, respectively. Then, based on the information bottleneck principle, we minimize semantic conditional entropy and maximize noise conditional entropy. This allows the SNN to retain se-

Table 8: Inference with reduced $T$ after training with $T = 4$.

| Method | $T = 1$ | $T = 2$ | $T = 3$ | $T = 4$ |
|---|---|---|---|---|
| Vanilla SNN | 10.0 | 60.1 | 72.7 | 72.9 |
| MPS Ding et al. (2025) | 24.1 | 66.9 | 73.0 | 75.8 |
| **Ours** | 16.5 | **69.3** | **76.8** | 78.2 |
| **Ours**+MPS | **38.2** | 67.3 | 76.4 | **78.6** |

mantics while ignoring noise and learn the optimal representation. Extensive experiments show that our method significantly improves the performance of low-latency SNNs. Furthermore, our method is highly generalizable and compatible with various architectures and methods. This provides fresh insights into semantic-noise decoupling in SNNs and low-latency, high-performance SNN training.

It is worth noting that our method employs an additional lightweight classifier to estimate conditional entropy during training, which introduces extra training overhead (albeit negligible). Further exploration of designing zero-overhead SNN training algorithms using efficient bit operations remains worthwhile.

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

---

**Algorithm 1** Ours entropy optimization algorithm for SNNs

---

**Require**: Input $X$, label $Y$, entropy optimization layer $\{layer_1, layer_2, \cdots, layer_l\}$, timestep $T$, balance coefficients $\gamma_{AND}$ and $\gamma_{XOR}$.
**Ensure**: Update SNN parameters

1: Forward propagation $X$ produces output $O$;
2: **if** Current layer in $\{layer_1, layer_2, \cdots, layer_l\}$ **then**
3:     Calculate $\hat{X}_{AND}$ and $\hat{X}_{XOR} \leftarrow$ Eq. 7;
4:     Calculate $O_{AND}$ and $O_{XOR} \leftarrow$ Eq. 9;
5:     Calculate $\mathcal{L}_{AND}$ and $\mathcal{L}_{XOR} \leftarrow$ Eq. 14;
6: **end if**
7: Aggregate losses across all layers;
8: Backpropagation for optimizing the SNN.

---

# A APPENDIX

## A.1 ANALYSIS OF AND/XOR OPERATIONS FOR DECOUPLING SEMANTICS/NOISE

In Fig. 1, we visualized spike maps to show that AND operations and XOR operations can decouple object semantics and spike noise, respectively. In this section, we further validate this conclusion with more in-depth analysis and quantitative experimental results.

We conclude that the AND operation decouples object semantics from spike maps, whereas the XOR operation primarily captures spike noise. The conclusion implies that the information decoupled by AND is more effective than the information decoupled by XOR. Accordingly, the accuracy of the AND operation output should significantly outperform that of the XOR operation when decoupled information is input into optimized subsequent layers to generate output. To this end, we conducted experiments on CIFAR10-DVS and DVS-Gesture. We decouple $\hat{X}_{AND}$ and $\hat{X}_{XOR}$ by performing AND and XOR operations on the final layer of the spike map in the optimized vanilla SNN. Then, we input both into the final classification layer to produce the output. Table 9 shows the comparative results of AND output and XOR output. The results show that the AND output is nearly as accurate as the original total output. In contrast, the XOR output is extremely inaccurate, indicating severe degradation. This illustrates that the information decoupled by the AND operation contains substantial object semantics, thereby achieving performance close to the original. In contrast, the information decoupled by the XOR operation is dominated by noise, resulting in degraded performance.

Table 9: Comparative results (%) of using total output and AND/XOR output. The AND output performs significantly better than the XOR output. This shows that the AND operation can produce enough object semantics, whereas the XOR operation produces results that are mostly noise.

| Dataset | Total output | AND output | XOR output |
|---------|-------------|------------|------------|
| CIFAR10-DVS | 72.9 | 71.9 | 22.7 |
| DVS-Gesture | 87.15 | 85.42 | 38.89 |

## A.2 THEORETICAL ANALYSIS OF EQUATION 8

In Eq. 8, we transform the minimization of $\mathcal{H}(Y|\hat{X})$ into the minimization of $\mathcal{H}(Y|\hat{X}_{AND})$ and the maximization of $\mathcal{H}(Y|\hat{X}_{XOR})$. Among them, $\hat{X}_{AND}$ is the object semantic and $\hat{X}_{XOR}$ is the spike noise. We assume that the information in $\hat{X}$ can be approximately decomposed into $\mathcal{H}(Y|\hat{X}_{AND})$ and $\mathcal{H}(Y|\hat{X}_{XOR})$:

$$\hat{X} \approx \{\hat{X}_{AND}, \hat{X}_{XOR}\}. \tag{16}$$

Ideally, $\hat{X}_{XOR}$ and $Y$ are independent of $\hat{X}_{AND}$:

$$Y \perp \hat{X}_{XOR} \mid \hat{X}_{AND}. \tag{17}$$

This means that given the object semantics $\hat{X}_{AND}$, spike noise $\hat{X}_{XOR}$ does not provide additional information about $Y$. At this point, the conditional entropy can be directly decomposed into:

$$\mathcal{H}(Y|\hat{X}_{AND}, \hat{X}_{XOR}) = \mathcal{H}(Y|\hat{X}_{AND}). \tag{18}$$

Therefore, minimizing $\mathcal{H}(Y|\hat{X})$ is nearly equivalent to minimizing $\mathcal{H}(Y|\hat{X}_{AND})$.

Note that if only $\mathcal{H}(Y|\hat{X}_{AND})$ is minimized, the SNN will still be affected by noise interference, similar to the standard training method of directly optimizing spike maps. Therefore, we need to prevent $\hat{X}_{XOR}$ from carrying information about the label. In other words, we need to explicitly degrade the amount of effective information in the spike noise. Formally, this requires minimizing the mutual information $I(\hat{X}_{XOR}; Y)$. This is equivalent to maximizing $\mathcal{H}(Y|\hat{X}_{XOR})$:

$$\begin{aligned} \min I(\hat{X}_{XOR}; Y) &= \min(\mathcal{H}(Y) - \mathcal{H}(Y|\hat{X}_{XOR})) \\ &= \min(-\mathcal{H}(Y|\hat{X}_{XOR})) \\ &= \max \mathcal{H}(Y|\hat{X}_{XOR}), \end{aligned} \tag{19}$$

Therefore, to minimize $\mathcal{H}(Y|\hat{X})$, we need to simultaneously minimize $\mathcal{H}(Y|\hat{X}_{AND})$ and maximize $\mathcal{H}(Y|\hat{X}_{XOR})$.

### A.3 EXPERIMENTAL DETAILS

**Neuromorphic Datasets.** We conducted experiments on three neuromorphic object recognition benchmark datasets: CIFAR10-DVS Li et al. (2017), DVS-Gesture Amir et al. (2017), and N-Caltech101 Orchard et al. (2015). The CIFAR10-DVS Li et al. (2017) dataset contains a total of 10,000 samples from ten categories. The DVS-Gesture Amir et al. (2017) dataset contains neuromorphic data for 11 hand gestures with 1176 training samples and 288 test samples. The spatial resolution of each sample in CIFAR10-DVS and DVS-Gesture is $128 \times 128$, which we downsampled to $48 \times 48$ and input to SNN, the standard input size for the community Ding et al. (2025); Zhou et al. (2024a). The N-Caltech101 Orchard et al. (2015) dataset contains 8,109 samples, categorized into 101 categories, with an original resolution of $180 \times 240$. We also downsampled the samples in N-Caltech101 to $48 \times 48$.

We conducted our experiments on an Ubuntu 20.04.5 operating system with an NVIDIA 4090 GPU. To validate the versatility of the proposed method, we conducted experiments using various architectures, including VGG-9, VGGSNN, and QKFormer. For VGG-9, we use the same training strategy as in Ding et al. (2025). The model was trained for 100 epochs using a stochastic gradient descent optimizer. The initial learning rate was set to 0.1, and it was scaled down tenfold every 30 epochs. The batch size is 64, and the weight decay is 1e-3. In particular, for VGG-9, we do not use any data augmentation during training. The firing threshold of the spiking neuron was set to 1.0, and the membrane potential time constant was set to 2.0. In the training of the SNN, the rectangular surrogate gradient function is employed for backpropagation, consistent with Ding et al. (2025).

When using VGGSNN for recognizing CIFAR10-DVS, we used random cropping and horizontal flipping as data augmentation. When using the VGGSNN architecture on the N-Caltech101 dataset, we incorporate the knowledge transfer strategy He et al. (2024) and employ the same training strategy as in He et al. (2024). Similarly, when using the QKFormer architecture, we follow the training strategy outlined in the original paper Zhou et al. (2024a) to ensure the fairness of the experiments.

We use the same architecture and experimental setup as in our method when we reproduce CLIF Huang et al. (2024), TAB Jiang et al. (2024), and SLT Anumasa et al. (2024), but we use the officially released core code implementation to ensure performance.

**Static Datasets.** For static images, we conducted experiments on the CIFAR10 and CIFAR100 datasets. Both the CIFAR-10 and CIFAR-100 datasets contain 60,000 $32 \times 32$ images, 50,000 of which are in the training set and 10,000 of which are in the test set. We normalized the CIFAR10 and CIFAR100 samples to a zero mean and unit variance. Then, we applied the standard data augmentation strategies, AutoAugment Cubuk et al. (2019) and Cutout DeVries & Taylor (2017).

For CIFAR10 and CIFAR100, we applied the ResNet-18 and ResNet-19 architectures. We used a stochastic gradient descent optimizer with a momentum of 0.9 to train for 300 epochs. The initial

Table 10: Structures of VGG-9, VGGSNN, ResNet-18, and ResNet-19, where fc denotes the fully connected layer.

| Stage | VGG-9 | VGGSNN | ResNet-18 | | ResNet-19 | |
|---|---|---|---|---|---|---|
| 1 | - | - | Conv($3 \times 3$@64) | | Conv($3 \times 3$@128) | |
| 1 | Conv($3 \times 3$@64) Conv($3 \times 3$@128) | Conv($3 \times 3$@64) Conv($3 \times 3$@128) | Conv($3 \times 3$@64) Conv($3 \times 3$@64) | $\times 2$ | Conv($3 \times 3$@128) Conv($3 \times 3$@128) | $\times 3$ |
| | AP(stride=2) | AP(stride=2) | - | | - | |
| 2 | Conv($3 \times 3$@256) Conv($3 \times 3$@256) | Conv($3 \times 3$@256) Conv($3 \times 3$@256) | Conv($3 \times 3$@128) Conv($3 \times 3$@128) | $\times 2$ | Conv($3 \times 3$@256) Conv($3 \times 3$@256) | $\times 3$ |
| | AP(stride=2) | AP(stride=2) | - | | - | |
| 3 | Conv($3 \times 3$@512) Conv($3 \times 3$@512) | Conv($3 \times 3$@512) Conv($3 \times 3$@512) | Conv($3 \times 3$@256) Conv($3 \times 3$@256) | $\times 2$ | Conv($3 \times 3$@512) Conv($3 \times 3$@512) | $\times 2$ |
| | AP(stride=2) | AP(stride=2) | - | | | |
| 4 | Conv($3 \times 3$@512) Conv($3 \times 3$@512) | Conv($3 \times 3$@512) Conv($3 \times 3$@512) | Conv($3 \times 3$@512) Conv($3 \times 3$@512) | $\times 2$ | | |
| | GAP,FC | AP,FC | GAP,FC | | GAP,FC$\times 2$ | |

learning rate was set to 0.1, and we employed a cosine annealing learning rate strategy. The batch size is 128, and weight deacy is 5e-4. Additionally, we use the QKFormer architecture to recognize static targets with the same training strategy as the original Zhou et al. (2024a).

On the ImageNet dataset, we train 300 epochs using ResNet-18 with a batch size of 256. The initial learning rate value is 0.1 and the learning rate is adjusted using a cosine annealing strategy. The default input size is $224 \times 224$, which is consistent with other methods.

Table 10 illustrates the architectures of the VGG-9, VGGSNN, ResNet-18, and ResNet-19 models that were used in the experiment. We performed entropy optimization on the last layer of the spike map for each stage when applying our method to these four architectures. For the QKFormer architecture, the CIFAR10-DVS and DVS-Gesture models contain two blocks, and entropy optimization is performed on the output spike maps of each block.

## A.4 IMAGENET EXPERIMENTS

Table 11 shows the comparative results on ImageNet. Due to limited computational resources, we only use the ResNet-18 architecture. For fairness, we compare our method with others that employ the same architecture. Under identical architecture and timestep settings, our method achieved an accuracy of 67.45%, significantly outperforming other methods. This demonstrates the effectiveness of our method on challenging large-scale datasets.

Table 11: Comparative results on large-scale ImageNet-1K.

| Method | Architecture | $T \downarrow$ | Accuracy (%) $\uparrow$ |
|---|---|---|---|
| ReverB-SNN Guo et al. (2025)[ICML'25] | ResNet-18 | 4 | 66.58 |
| IMP+LTS Shen et al. (2024a)[NeurIPS'24] | ResNet-18 | 4 | 65.38 |
| EnOF-SNN Guo et al. (2024)[NeurIPS'24] | ResNet-18 | 4 | 65.31 |
| AGMM Liang et al. (2025)[AAAI'25] | ResNet-18 | 4 | 64.67 |
| ETC Zhao et al. (2025)[PR'25] | ResNet-18 | 4 | 63.70 |
| KDSNN Xu et al. (2023a)[CVPR'23] | ResNet-18 | 4 | 63.42 |
| SEW-ResNet Fang et al. (2021)[NeurIPS'21] | ResNet-18 | 4 | 63.18 |
| MPBN Guo et al. (2023b)[ICCV'23] | ResNet-18 | 4 | 63.14 |
| RMP-Loss Guo et al. (2023a)[ICCV'23] | ResNet-18 | 4 | 63.03 |
| SSCL Zhang et al. (2024)[AAAI'24] | ResNet-18 | 4 | 62.95 |
| **Ours** | ResNet-18 | 4 | **67.45** |

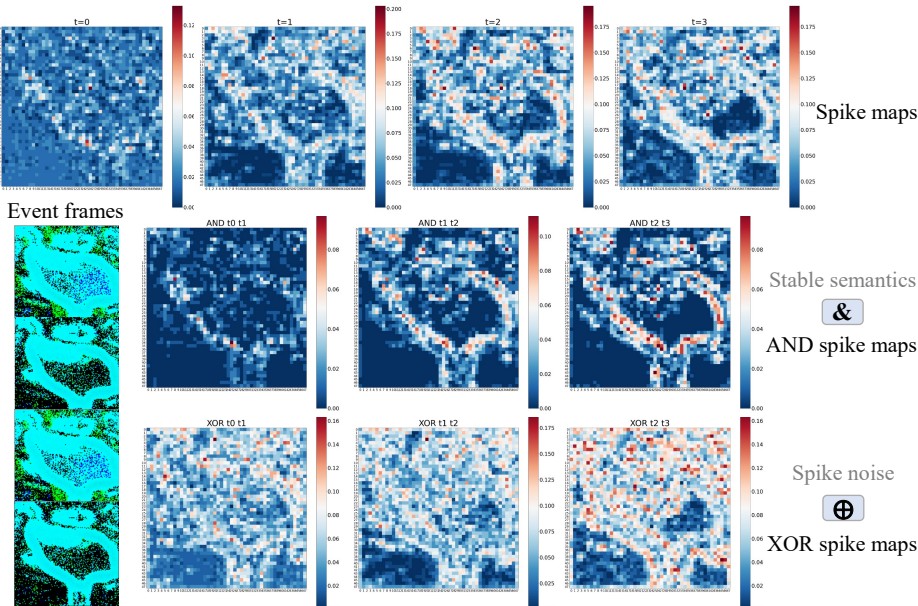

Figure 4: Visualization of spike maps generated by AND and XOR bit operations on the CIFAR-10 DVS dataset (second layer of spiking VGG-9). Compared to the original vanilla spike maps (first row), the AND spike maps (second row) extract well-defined object semantics, while the XOR spike maps (third row) capture spike noise.

## A.5 ADDITIONAL VISUALIZATIONS

To demonstrate the prevalence of spike noise in spike maps and the effectiveness of AND and XOR operations more clearly, we present additional spike map visualizations in Figures 4 through 10. These results suggest that spike noise is prevalent across datasets and layer depths. They also indicate that AND and XOR operations consistently decouple object semantics from spike noise effectively and efficiently.

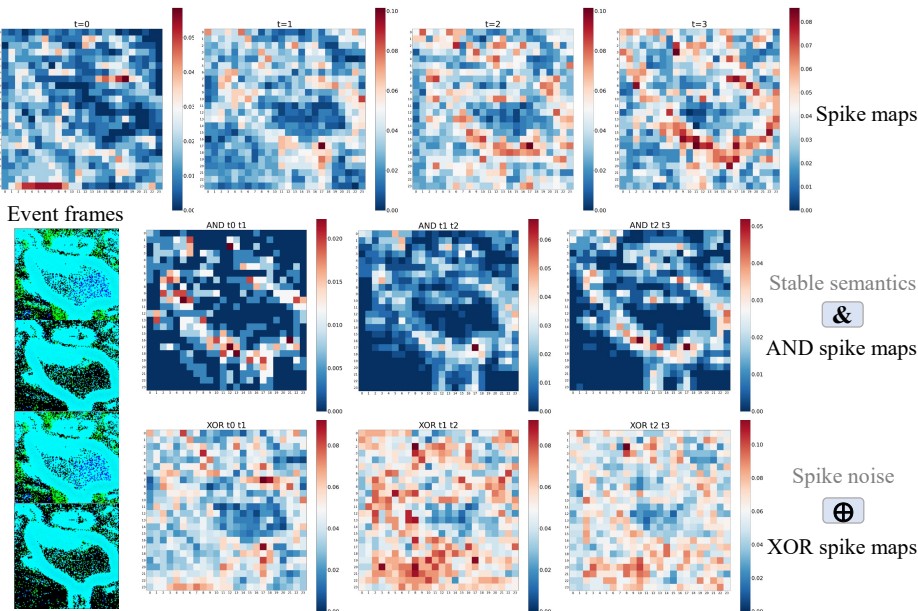

Figure 5: Visualization of spike maps generated by AND and XOR bit operations on the CIFAR-10 DVS dataset (third layer of spiking VGG-9). Compared to the original vanilla spike maps (first row), the AND spike maps (second row) extract well-defined object semantics, while the XOR spike maps (third row) capture spike noise.

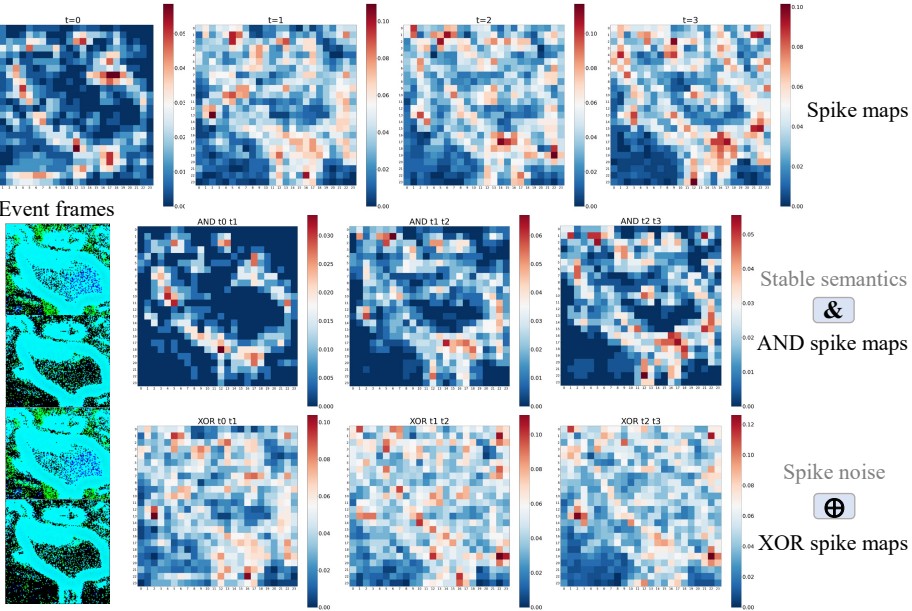

Figure 6: Visualization of spike maps generated by AND and XOR bit operations on the CIFAR-10 DVS dataset (fourth layer of spiking VGG-9). Compared to the original vanilla spike maps (first row), the AND spike maps (second row) extract well-defined object semantics, while the XOR spike maps (third row) capture spike noise.

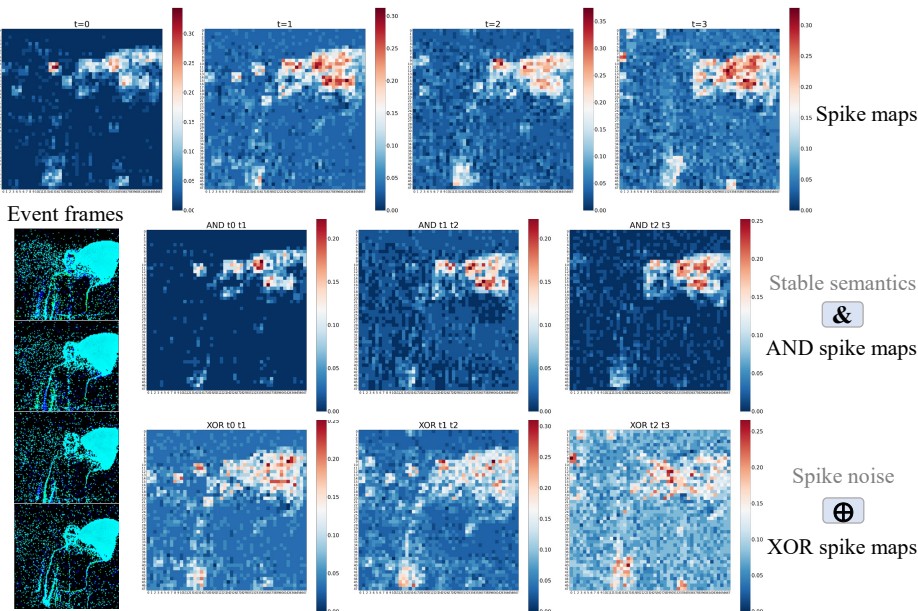

Figure 7: Visualization of spike maps generated by AND and XOR bit operations on the DVS-Gesture dataset (first layer of spiking VGG-9). Compared to the original vanilla spike maps (first row), the AND spike maps (second row) extract well-defined object semantics, while the XOR spike maps (third row) capture spike noise.

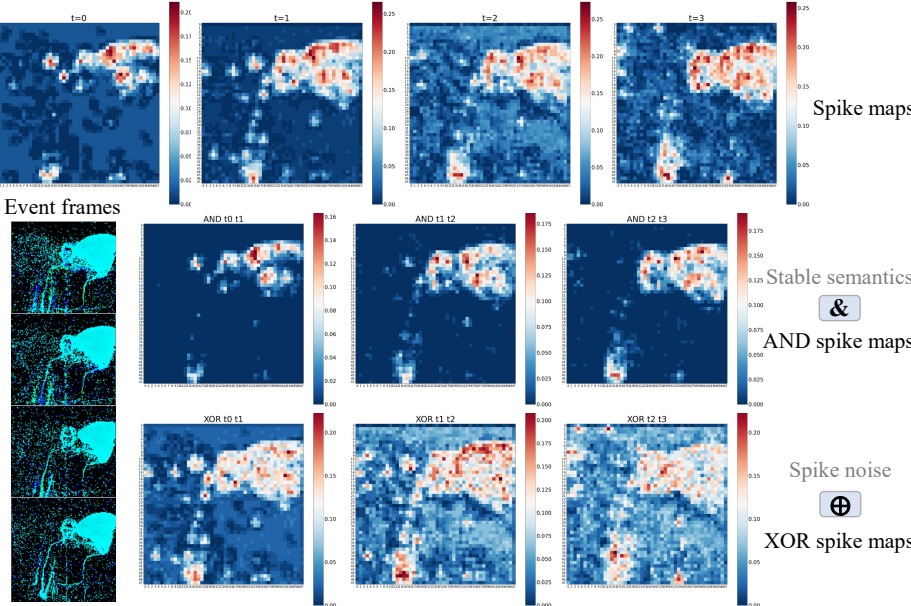

Figure 8: Visualization of spike maps generated by AND and XOR bit operations on the DVS-Gesture dataset (second layer of spiking VGG-9). Compared to the original vanilla spike maps (first row), the AND spike maps (second row) extract well-defined object semantics, while the XOR spike maps (third row) capture spike noise.

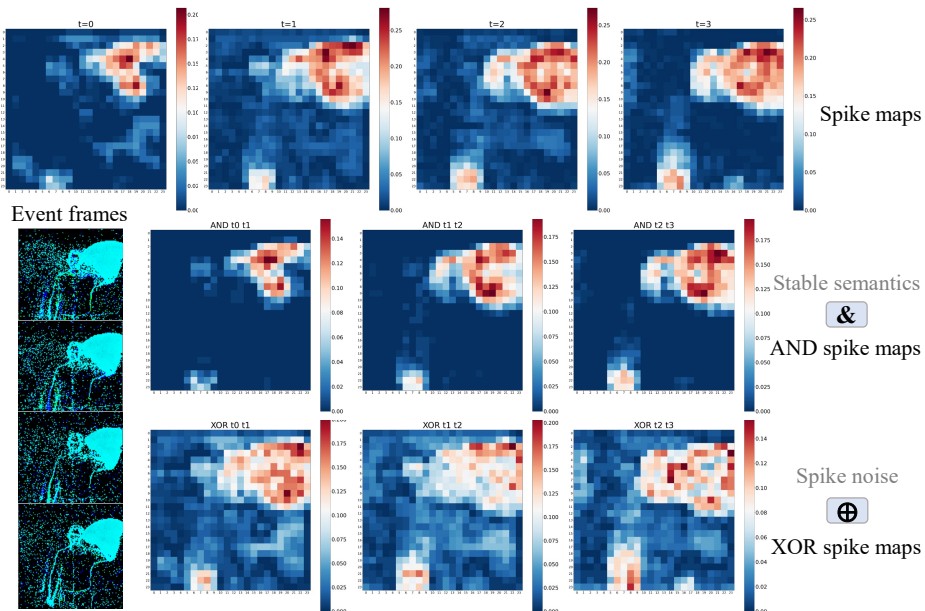

Figure 9: Visualization of spike maps generated by AND and XOR bit operations on the DVS-Gesture dataset (third layer of spiking VGG-9). Compared to the original vanilla spike maps (first row), the AND spike maps (second row) extract well-defined object semantics, while the XOR spike maps (third row) capture spike noise.

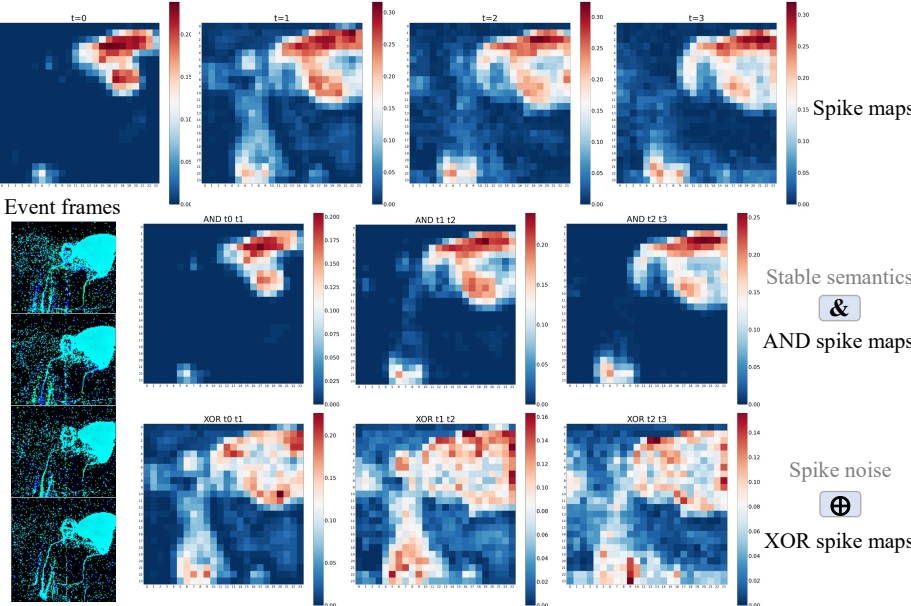

Figure 10: Visualization of spike maps generated by AND and XOR bit operations on the DVS-Gesture dataset (fourth layer of spiking VGG-9). Compared to the original vanilla spike maps (first row), the AND spike maps (second row) extract well-defined object semantics, while the XOR spike maps (third row) capture spike noise.

