# OpenReview forum: "From Noise to Semantics: Bit Operations Decouple Spiking Neural Networks for Entropy Optimization"
_ICLR.cc/2026/Conference — ICLR 2026 Conference Withdrawn Submission_

### Official Review · Reviewer_U7yB · 2025-10-30

**Soundness:** 2
**Presentation:** 3
**Contribution:** 2
**Rating:** 4
**Confidence:** 3

**Summary:**

The paper proposes a lightweight classifier and a training loss for SNN BPTT training. The loss consists of three parts: the cross-entropy loss between the target and the SNN output, the cross-entropy loss between the X after nor operation and target, and the cross-entropy loss between X after and operation and the target. By maintaining the semantics in stable activation $X_{and}$ and maximizing the semantics in spike noise $X_{nor}$, the work achieves state-of-the-art accuracy in CIFAR10-DVS datasets.

**Strengths:**

1. Paper writing is good and smooth. The motivation of the training loss makes sense.
2. The experiment results in dynamic datasets are impressive.
3. The formulation derivation is clear.

**Weaknesses:**

1. Lack of training in overhead analysis

The paper claims that the proposed method introduces minimal additional training cost. However, only the increase in parameter count and FLOPs is reported. The actual GPU memory usage and training time are not provided, which are important indicators of training efficiency. The reason is that additional bitwise operations (e.g., XOR and AND for feature computation) may also incur non-negligible computational overhead that should be discussed or quantified.

2. Limited experimental datasets

The experiments are primarily conducted on relatively simple datasets. The most challenging dataset is CIFAR-10-DVS. To better validate the scalability and generalization capability of the proposed approach, results on more complex datasets, such as HAR-DVS and N-ImageNet, would be highly valuable.

3. The position of for XOR/AND features

The paper does not clearly specify where the XOR and AND operations are performed in the network. For instance, in the case of VGG9, are these operations applied to the feature maps before the head classifiers? What is the motivation for choosing this particular location?

**Questions:**

Please see the weaknesses.

---

### Official Review · Reviewer_5JdX · 2025-10-30

**Soundness:** 1
**Presentation:** 2
**Contribution:** 1
**Rating:** 2
**Confidence:** 5

**Summary:**

This paper attempts to decouple semantic and noisy information in spiking neural networks (SNNs) via simple bitwise operations (AND/XOR) combined with the Information Bottleneck (IB) principle. Although the idea appears novel, the work suffers from major weaknesses in theoretical soundness, experimental rigor, and substantive innovation. The proposed method is largely heuristic and lacks sufficient analytical justification or fair comparison. Therefore, I believe the paper does not meet the acceptance standards of ICLR.

**Strengths:**

The paper identifies an important challenge in spiking neural networks (SNNs) — the interference of spike noise with object semantics — and proposes to tackle it from an information-theoretic perspective. The idea of connecting minimal bitwise operations with entropy optimization is conceptually creative. The use of AND/XOR operations aligns naturally with binary spike representations and can be efficiently implemented on neuromorphic hardware. This could inspire future studies on ultra-low-power SNN algorithms.

**Weaknesses:**

1. The core assumption — that XOR features are conditionally independent of the label Y given AND features — is extremely strong and unsupported. Given the nonlinear and temporally dependent nature of SNNs, this conditional independence is unrealistic. Consequently, the theoretical derivation of minimizing/maximizing conditional entropy is built on fragile reasoning.
2. The so-called “bit operation decoupling” essentially performs pixel-wise temporal comparisons, which is equivalent to fixed-weight convolution or temporal consistency filtering. While intuitive, this does not constitute a substantial algorithmic or conceptual innovation.
3.  Although the paper claims to employ the IB principle, the implementation only estimates output entropy via a lightweight classifier: Mutual information is never computed or constrained. No variational inference or proper optimization is applied. There is no theoretical guarantee that the approximation aligns with the IB objective. Thus, “entropy optimization” functions more as a heuristic regularizer than a rigorous information-theoretic framework.
4. Tables 5 and 6 compare many methods without clarifying whether training epochs, hyperparameters, and augmentations are consistent. The reported gains (1–2%) lack standard deviation or confidence intervals, so statistical significance cannot be assessed. The qualitative visualizations rely on color contrast and subjective interpretation, without quantitative noise metrics.
5. The paper frequently asserts “significant improvement” and “strong generalization,” but the actual gains are small and verified only on limited datasets. Moreover, the claimed “low-latency” inference is not clearly supported — the method introduces extra training losses but does not reduce inference timesteps.
6. The paper claims that AND and XOR operations can decouple semantics and noise, but there is no explanation of how these operations interact with the gradient-based learning process. Although surrogate gradients (Eq. 15) are introduced, they are extremely crude approximations that could easily distort the optimization landscape. The paper fails to show whether this approximation maintains convergence stability or whether it introduces gradient bias that harms learning. A theoretical or empirical analysis of the optimization behavior is entirely missing.
7. The proposed entropy-based regularization should be compared with simpler, established alternatives such as dropout, noise injection, or consistency regularization across timesteps. Without such baselines, it is impossible to determine whether the improvement stems from the proposed entropy framework or simply from the extra regularization effect of the auxiliary loss.

**Questions:**

Please see my concerns on weakness.
In addition, other questions are listed as follows:
1. Did you observe any gradient instability or saturation effects during training?
2. Why were these specific forms chosen instead of smoother differentiable approximations?
3. Have you empirically measured the mutual information to confirm the claim?
4. How does it compare empirically with simpler regularization techniques such as dropout, noise injection, or temporal consistency losses?

---

### Official Review · Reviewer_SaRa · 2025-10-31

**Soundness:** 2
**Presentation:** 3
**Contribution:** 2
**Rating:** 4
**Confidence:** 4

**Summary:**

This paper proposes a novel optimization method for SNNs. The core idea is to decouple the spike graphs of adjacent time steps by using AND and XOR, two lightweight bit operations, to separate "object semantics" from "spike noise". Subsequently, based on the principle of information bottleneck, the authors design an entropy optimization strategy, aiming to minimize the conditional entropy of semantic features while maximizing the conditional entropy of noise features, thereby guiding the network to learn more robust feature representations.

**Strengths:**

1. The proposed method introduces minimal overhead during the training phase and does not increase any inference cost, which is a significant advantage for the practical deployment of SNNs. It fully exploits the binary nature of spikes, and the method itself is hardware-friendly and highly promising.

2. The experiment is thorough, covering various architectures, datasets, and low-latency scenarios. Particularly, the experiments on the collaborative work with mechanisms such as early exit demonstrate the good compatibility and practical value of the method.

**Weaknesses:**

1. The paper claims that the "stable firing across time steps" captured by the AND operation represents object semantics, but this is inherently contradictory to the reset mechanism of LIF neurons after firing. It is difficult for the same neuron to fire again immediately in the next time step after reset. The paper fails to explain what the microscopic basis of this "stability" is, which makes its core premise seem insufficiently rigorous.

2. The information bottleneck principle is used to explain entropy optimization, but the division of "semantics" and "noise" completely relies on the heuristic method of AND/XOR operations. This division lacks strict proof and seems more like an effective "engineering trick" that was later packaged with theory rather than a method naturally derived from theory.

3. The real reason for the method's effectiveness may not be "extracting stable semantics over time", but rather achieving efficient spatial feature selection and filtering through AND/XOR. The paper fails to deeply explore alternative explanations for its effectiveness, and the depth of discussion is insufficient.

4. Limited performance improvement: For example, on DVS-Gesture dataset, compared with the baseline QKFormer at T=16, the accuracy is not improved.

**Questions:**

1. Given the reset mechanism of LIF neurons, what is the specific microscopic mechanism at the neuronal level for the "continuous firing" captured by the AND operation? Is it caused by an abnormal strong drive of the same neuron, or is it a macroscopic statistical result of different neuronal populations at the same spatial location?

2. If it is mainly a group statistical effect (i.e., different neurons fire at the same location at time t and t+1), does this weaken the argument of "time stability"? Does this mean that this method is more like an attention mechanism based on spatial consistency rather than a true time dynamic analysis?

3. Can the author provide more microscopic evidence (for example, tracking the changes in the membrane potential and input current of the neurons contributing to the strong AND signal over time) to directly prove that its "stable semantics" indeed stems from overcoming the reset effect and the continuous strong input drive, thereby supporting its core argument?

---

### Official Review · Reviewer_rMSJ · 2025-11-01

**Soundness:** 2
**Presentation:** 3
**Contribution:** 3
**Rating:** 4
**Confidence:** 4

**Summary:**

This paper introduces a novel method for improving the performance of spiking neural networks (SNNs) by efficiently decoupling stable object semantics from spike noise using minimal bit operations (AND and XOR). The proposed information bottleneck-based entropy optimization strategy minimizes the conditional entropy of object semantics while maximizing the entropy of spike noise, allowing SNNs to focus on meaningful representations. Extensive experiments demonstrate that this approach significantly enhances SNN performance, enabling flexible and ultra-low latency inference with strong generalization capabilities. The method also integrates seamlessly with other techniques, offering potential for broader applications in SNN-based systems.

**Strengths:**

1) The proposed minimal bit operation decoupling method effectively reduces spike noise in Spiking Neural Networks (SNNs), improving semantic representation accuracy and performance.

2) By combining the information bottleneck principle with entropy optimization, the method significantly reduces noise interference during training, enhancing the generalization ability of SNNs.

3) The approach introduces minimal training overhead while being compatible with other techniques, enabling flexible timestep inference and ultra-low latency early exit mechanisms.

**Weaknesses:**

1) The training dataset is too small, such as CIFAR. So, it is hard to evaluate the scalability of this method.

2) Based on the first point, this work adds some loss information to the spiking feature, which I think is not a good way to train very large spiking models. The training stability is not addressed. More neuronal dynamics could be explored.

**Questions:**

See weaknesses.

---

### Note · Authors · 2025-11-23

I have read and agree with the venue's withdrawal policy on behalf of myself and my co-authors.